# Perspectives of women, healthcare providers and health managers on Group Antenatal Care implementation in Geita, Tanzania: A qualitative study

Alen Kinyina[1,2]*, Augustino Hellar[1], Raymond Bandio[1], Hamid Mandali[1], Ahmad Makuwani[3], Abubakari Munga[1], Phineas Sospeter[3], Yusuph Kulindwa[1], Ntuli Kapologwe[4], Frank Phiri[1], Isaac Lyatuu[1], Wilfred Kafuku[1], Cyprian Mtani[1], Husna Athumani[1], James Hellar[3], Omari Sukari[5], James Tumaini Kengia[6,7], Erin Jones-Avni[2], Hermes Gichane[2]

**1** Prime Health Initiative Tanzania (PHIT) Dar es Salaam, Tanzania, **2** London School of Hygiene and Tropical Medicine (LSHTM), London, United Kingdom, **3** Ministry of Health, Dodoma, Tanzania, **4** East, Central, and Southern Africa Health Community (ECSA-HC), Arusha, Tanzania, **5** Geita Regional Health Management Team (RHMT), Geita, Tanzania, **6** President's Office Regional Administration and Local Government (PORALG), Dodoma, Tanzania, **7** University of Dodoma, Dodoma, Tanzania

* alenkinyina@gmail.com

## Abstract

### Background

Among the root causes of high maternal and neonatal mortality is low utilization of antenatal care (ANC) services and sub-optimal quality of care during pregnancy. Group Antenatal Care (G-ANC) is an innovative model of care where pregnant women with similar gestational age (GA) meet for their antenatal care visits as a group, rather than individually. This approach promotes peer support, shared learning, and a sense of community among pregnant women, and enhance the quality of care. This study aimed to assess the perspectives of healthcare providers (HCPs), women and health managers to determine the experience and acceptability of G-ANC in low resource settings.

### Methods

The project was implemented from February 2023 to June 2024. However, participant recruitment for the qualitative component commenced in June to September 2024. Data was collected through six Focus Group Discussions (FGDs) with pregnant women and postnatal women who had recently attended G-ANC sessions. Key Informant Interviews (KIIs) were conducted with HCPs, facility leaders (in-charge) and district / regional health managers. Respondents were selected through purposive sampling, and discussions were conducted using semi-structured interview guides.

**Data availability statement:** There is no restriction on access to the de-identified dataset generated by this study and can be accessed at: https://doi.org/10.7910/DVN/RLAJMC. For the qualitative component, no personal identifiable information was collected, and in some cases, interviews were de-identified at the point of transcription. Interested researchers may seek the access approval from the the National Health Research Ethics Committee (NatHREC) via the secretariat at ethics@nimr.or.tz.

**Funding:** "This project was funded by the Gates Foundation under grant number INV-046249 (https://gcgh.grandchallenges.org/grant/mlinde-mama-digital-health-platform). Additional information about the Foundation is available at https://www.gatesfoundation.org. The funder had no role in the study design, data collection, analysis, interpretation or preparation of the manuscript. In accordance with grant requirements, the funder reviewed and approved the manuscript for publication but did not influence the content or conclusions".

**Competing interests:** The authors have declared that no competing interests exist.

Audio recorded interviews were transcribed, and thematic analysis was conducted using NVIVO software.

## Results

Findings revealed positive experience, satisfaction and acceptability of G-among women and HCPs. Women reported increased knowledge on maternal health, emotional support, and enhanced communication with HCPs. Healthcare providers highlighted improved efficiency, better patient engagement, and peer support among clients. Health managers demonstrated contribution of G-ANC in promoting increased ANC attendance and more interactive care. However, challenges such as limited space, need for facilitator training, and resource constraints were identified.

## Conclusion

Group Antenatal Care demonstrated significant benefits for maternal health education, peer support, and healthcare delivery efficiency. These findings add evidence to other existing literature supporting the adoption of this intervention in low resource settings. However, successful implementation requires investments in infrastructure (space for G-ANC meetings), adequate and trained HCPs, service integration and male engagement. Institutionalizing G-ANC as a standard antenatal care could be the best option to address the low coverage of ANC visits, suboptimal quality care, and promotion of institutional deliveries in low resource settings.

## Background

Maternal and neonatal mortality remain a pressing issue worldwide. Every two minutes, a woman dies from complications related to pregnancy or childbirth, and each day, 6,300 newborns die [1]. Every year about 2.3 million newborns and 303,000 women die due to pregnancy or childbirth [1]. According to the World Health Organisation (WHO), approximately more than 70% of maternal deaths occur in sub-Saharan Africa [2]. In Tanzania, maternal mortality ratio is 104 per 100,000 live births which is higher than the SDGs target of 70 deaths per 100,000 live births [3]. To improve the outcomes of pregnancy and newborn health, the WHO recommends adequate adherence to antenatal care (ANC) [4].

Antenatal care refers to "care provided by skilled healthcare professionals to pregnant women and adolescent girls to ensure the best health conditions for both mother and baby during pregnancy" [4]. Adequate ANC services, including routine screening and diagnosis, reduce the frequency of maternal death, miscarriages, birth defects, low birth weight, neonatal infections, and other preventable health problems [4]. Workload, lack of essential commodities, and demotivated HCPs lead to low satisfaction of services and poor utilisation of ANC services among pregnant women in countries with limited resources [4,5].

In 2016, the WHO revised the ANC guideline, recommending each pregnant woman attend at least eight contacts instead of four visits [6]. However, the coverage of recommended ANC visits and the provision of quality ANC services is still low in many developing countries. In Tanzania, the ANC coverage of at least four visits is less than 68% [7,8]. As such, various approaches have been taken to ensure high coverage of ANC visits, including the implementation of the innovative G-ANC [9,10]. Group Antenatal Care was originally implemented in high-resource settings and has been found to have a beneficial effect on pregnancy care, outcomes, high acceptability and positive experience of care [11]. In G-ANC women receive antenatal services in the group at the same time during group visits, as well as creating a support group of women at a similar stage in pregnancy. During G-ANC, a group of 8–15 women with similar GA meet for ANC visits which include self-assessments, individual private clinical ANC consultations with the HCPs, and a facilitated, highly participatory group session to learn, share, and problem solve [12,13].

Group Antenatal Care has the potential to be more efficient, as the provider can provide information relevant to the stage of pregnancy to all women at the same time rather than to each woman individually [12,14,15]. A key part of G-ANC is that women do some of their own assessments (blood pressure, weight), with the provider as a facilitator, to encourage their investment in self-care and care of the foetus [5,15]. Additionally, because the women are all at a similar stage of pregnancy, the GANC sessions develop a peer support network where women are receiving the same health information and sharing many of the same pregnancy experiences [12,15].

Previous studies from various low- and middle-income country settings demonstrates that G-ANC is highly acceptable to women and HCPs. Acceptability is defined as the extent to which an intervention perceived as appropriate, satisfactory, and align with stakeholders' expectations and values [16]. High preference of G-ANC among women is driven by opportunities for peer support, shared learning, and improved communication with HCPs. Studies from diverse contexts, including Kenya [13,12], Nigeria [13,12], Ghana [5], Rwanda [17], Nepal [18], and Bangladesh [19], report that women prefer the group care for enhancing understanding of pregnancy-related health information and increasing confidence in engaging with care. Another, systematic review demonstrated that G-ANC is associated with improving women's satisfaction with antenatal services and perceived quality of care [20]. Acceptability of G-ANC has also been documented among healthcare providers. A systematic review conducted by Lazar et al. (2021) highlights that providers across various settings generally view G-ANC as valuable approach for improving ANC services. HCPs noted the link between G-ANC with improved efficiency, more client engagement and enhanced provider–client relationships.

Because of increased interest in adaptation of G-ANC within low-resource settings, donors and policymakers have turned attention to this ANC service delivery model [11,12,15]. In Tanzania, the first study on G-ANC was conducted in 2014 as a randomized controlled trial in Dar es Salaam [21]. This study reported positive effects on women's engagement with antenatal services and selected maternal health outcomes. Another study was conducted in Kibaha district exploring perceptions of women and midwives on ANC information and communication during group care [22]. In addition, G-ANC has been implemented programmatically in Zanzibar, although findings from that context have not yet been formally published. However, these studies were conducted in only urban settings and focused primarily on clinical and behavioural outcomes with limited assessments of stakeholder experiences and acceptability assessments.

In 2022, a pilot implementation project was introduced in Tanzania to assess the effect of G-ANC on uptake of antenatal services [23]. A total of 5,936 women across 149 cohorts were enrolled. Among them, 93.9% completed four or more ANC visits, 92.6% received iron–folate supplementation, and 96.2% delivered at health facilities [23]. Findings showed that G-ANC was associated with increased proportion of women completing four or more ANC visits from 34%recorded during the baseline to 93.9%. The present qualitative study stems from this pilot project. Other previous studies in Tanzania indicated that women attending prenatal care through the G-ANC demonstrated a high level of engagement, perceived quality of care and retention in care [9,24]. This highlights a significant opportunity and the potential of adopting this intervention in routine ANC practices to enhance ANC coverage, quality care, and ultimately reduce maternal and neonatal mortality.

 

While G-ANC has been previously studied in Tanzania, existing evidence is limited in scope and context. As a result, there remains a critical evidence gap regarding how women, healthcare providers, and health managers experience G-ANC, particularly in no-urban regions contexts like Geita. Limited understanding of stakeholder perspectives may constrain policy-maker confidence, reduce stakeholders' ownership, and undermine successful institutionalization of new interventions like G-ANC. Therefore, this study aimed to explore stakeholders' experiences with G-ANC and to explore its acceptability within the low resource settings. If validated, the G-ANC can augment or replace the current conventional ANC approach.

## Methodology

### Qualitative study design

Given the exploratory nature of this research, the study design followed a qualitative approach. Qualitative study design is an appropriate method in assessing experience, views and perspectives about a phenomenon [25]. Focused group discussions (FGDs) were conducted with women who attended G-ANC and KIIs conducted with the HCPs and health managers. The FGD and KII guides were used in gathering views and perspectives of the respondents on a variety of topics about G-ANC implementations in their settings. Modified interview guides were adopted from elsewhere [5]. The project was implemented from February 2023 to June 2024. However, participant recruitment for the qualitative component commenced in June 2024 to August 2024. The consolidated criteria for reporting qualitative research (COREQ) checklist was used to enhance the rigour and transparency of the study's reporting (SI).

### Selection of participants

Pregnant women or recently delivered women form six facilities were purposefully selected for participation in FGDs. Facilities were categorized as hospital-level, health center-level, and dispensary-level. Two FGDs were conducted with pregnant women and four FGDs with mothers who delivered within the past 6 months from the selected facilities' catchment areas until we reached saturation in the responses. The snowball sampling method was used to reach women who participated in G-ANC previously. Each FGD consisted of at most 8–12 women. KIIs were conducted with at least one health care provider and the facility leader from each facility. HCPs included doctor/ assistant medical officer, clinical officer, assistant clinical officer, and nurse/midwife. Traditional birth attendants and village health workers were not included in the definition of HCPs. Health Managers included the district/regional maternal health managers or coordinators.

### Sample size determination and triangulation

Sampling and sample closure were guided by the principles of purposive sampling, triangulation, and thematic saturation, rather than by a predetermined sample size. Participants were selected to ensure representation of key stakeholder groups involved in the design, delivery, and receipt of G-ANC, namely women receiving care, healthcare providers, and health system managers. Sample closure was reached when thematic saturation was achieved, defined as the point at which additional FGDs or KIIs no longer yielded new concepts, or substantive insights relevant to the study objectives. Saturation was assessed separately within each participant group and then across groups. Sample triangulation was achieved through the inclusion of multiple perspectives across different roles (service users, providers, and managers), facility levels (hospital, health centre, and dispensary), and geographical contexts (urban, peri-urban, and rural settings). This triangulated sampling strategy allowed for comparison and validation of findings across participant categories and service delivery contexts.

### Data collection and procedures

Data were collected through FGDs with pregnant and postnatal women who had participated in G-ANC and KIIs with antenatal HCPs and health managers. Semi-structured interview guides were used for both FGDs and KIIs. The core thematic areas explored during FGDs and KIIs included:

1. Perceptions and experiences of G-ANC

2. Practicality within routine health services

3. Overall acceptability of G-ANC among women, healthcare providers, and health managers.

4. Challenges to successful implementation of G-ANC

Examples of guiding (trigger) questions included: *"Can you describe your experience attending or providing care through G-ANC?"*, *"What aspects of G-ANC worked well, and what challenges were encountered?"*, and *"How does G-ANC fit within routine antenatal care services at this facility?"* Probing questions were used flexibly to explore emerging issues in greater depth.

FGDs and KIIs were conducted in Kiswahili by trained qualitative researchers who were not involved in service delivery. All discussions were held in private, quiet locations within or near health facilities to ensure confidentiality and participant comfort. Interviews and discussions were audio-recorded with participant permission, and detailed field notes were taken to capture non-verbal communication and contextual information.

Each FGD was conducted by a trained facilitation team, consisting of a lead facilitator (moderator) and a note-taker (observer). The facilitator guided the discussion, ensured balanced participation, and followed the interview guide, while the note-taker documented key points, group dynamics, and non-verbal cues and managed logistical aspects, including audio recording. In some sessions, an additional research team member provided logistical support. Although a separate co-facilitator role was not formally designated for all FGDs, the note-taker actively supported the facilitator in managing group flow and ensuring all participants had an opportunity to contribute.

## Data analysis

Audio recordings were transcribed verbatim and translated from Kiswahili into English by trained bilingual researchers. Transcripts were reviewed against audio files to ensure accuracy. Data were imported into NVivo software for coding and thematic analysis. Analysis followed a hybrid deductive–inductive approach. Coding was conducted iteratively by multiple members of the research team, with regular discussions to compare interpretations, resolve discrepancies, and refine themes.

To enhance credibility and dependability, methodological triangulation was achieved by comparing perspectives across women, healthcare providers, and health managers, as well as across FGDs and KIIs. Data collection continued until thematic saturation was reached. Transferability was supported through detailed description of the study context, participant characteristics, and implementation setting (all facilities).

## Ethical considerations

The ethical approval for this project was sought from the National Institute of Medical Research (NIMR) Tanzania, under approval certificate number NIMR/HQ/R.8a/Vol.IX/4194 dated January 23rd, 2023. The team considered all necessary measures to ensure the recruitment, consenting, data collection, and publication of the materials related to this study are following regulations governing research standards. Key consenting processes were taken to ensure voluntary participation of the study participants, protection of their privacy, confidentiality of the data and their safety.

## Informed consent

Verbal consent was obtained from women prior to their engagement in G-ANC services, aligning with the standard procedures commonly followed during conventional antenatal care. Healthcare providers explained the G-ANC, its purpose, and what participation entailed during ANC visits. Women's verbal consent to participate was recorded in facility ANC

registers and cohort tracking tools, consistent with routine service documentation practices. This protocol was reviewed and approved by the Institutional Review Board (IRB).

For participants involved in the qualitative component of the study to explore their perceptions and experiences, written consents were obtained prior to conducting the interviews. The consenting process was administered by trained research staff and conducted in Kiswahili (the national language) to ensure comprehension and cultural appropriateness. Each participant signed two copies of the consent form: one was retained by the participant for their records, and the second was securely stored by the study team. The consent forms included components outlining the purpose of the study, procedures involved, potential risks and benefits, confidentiality measures, voluntary participation, and the right to withdraw at any time without affecting their access to healthcare services. Participants were encouraged to ask questions, and their understanding was verified prior to signing the forms.

## Results

Twenty-one KIIs were conducted with ANC providers, facility in-charges from all six facilities, and three health managers. A total of six FGDs were conducted, including four with postnatal women and two with pregnant women who were continuing with G-ANC (Table 1). The views and perspectives of the various participants were organized into themes, as presented in this section.

### Experience of women about Group Antenatal Care (G-ANC)

Women who participated in the qualitative study reported positive experiences, emphasizing the benefits of shared learning, emotional support, and improved access to healthcare services. Group care was recognized as an effective platform for empowering women with essential maternal health knowledge. Women expressed that attending G-ANC sessions helped them gain a deeper understanding of pregnancy-related issues, including nutrition, birth preparedness, and newborn care. One participant shared,

**Table 1. Characteristics of study participants.**

| Participant category Women (FGDs) | Characteristics | Number of respondents (n) |
|---|---|---|
| | Pregnant women | 18 |
| | Recently delivered women (≤6 months) | 35 |
| | **Facility/catchment area** | |
| | Nkome Dispensary | 12 |
| | Butengorumasa Dispensary | 9 |
| | Chato District Hospital | 13 |
| | Nzera District Hospital | 12 |
| | Bwanga Health Center | 15 |
| | Katoro Health Center | 13 |
| **Healthcare providers (KIIs)** | Medical doctors | 3 |
| | Clinical officers/assistant clinical officers | 4 |
| | Nurses/midwives | 5 |
| | Facility in-charges | 6 |
| **Health managers (KIIs)** | District health coordinators | 2 |
| | Regional Health Coordinator | 1 |

*"Before joining the group sessions, I was not aware of the importance of balanced nutrition during pregnancy. Now, I not only eat better but also encourage other women in my community to do the same.."* Postnatal woman, Health Centre 2

Pregnant women who were still attending G-ANC similarly described improvements in knowledge and birth preparedness.

*"Since joining the group, I have learned many things about how to take care of myself and the baby. I now understand what to prepare before delivery, which I did not know before."* Pregnant woman, Dispensary 1.

The interactive nature of G-ANC also created a sense of community among the women, reducing feelings of isolation during pregnancy. Some women appreciated the opportunity to share experiences and learn from one another.

*"It felt reassuring to be in a group with other pregnant women who were going through the same journey as me...We encouraged each other and became like a family."* Postnatal woman, District Hospital 2

Women emphasized how G-ANC fosters a sense of belonging, emotional support, and collective learning, making pregnancy a more positive and informed experience for women. This support system played an important role in alleviating fears and anxieties related to pregnancy and childbirth as noted from some women who said:

*"Listening to other women's experiences made me realize that I was not alone in my struggles. We shared advice, and it helped me feel more confident about my pregnancy."* Postnatal woman, Dispensary 2

*"In the group, we laughed, learned, and supported each other. It was not just about medical check-ups; it was a place where we felt heard and understood."* Pregnant woman, Health Centre 2

*"I used to feel anxious about childbirth, but hearing stories from other mothers who had gone through it helped ease my fears. Their experiences gave me strength and encouragement."* Postnatal woman, Health Centre 1

Additionally, women noted that G-ANC improved their confidence in communicating with healthcare providers. This increased engagement contributed to better adherence to recommended antenatal care practices. They felt more comfortable asking questions and seeking clarification about their health. One participant explained,

*"In the past, I was afraid to ask the nurse questions, but in the group, we openly discuss everything, and I feel more prepared for delivery."* Postnatal woman, District Hospital 1

*"Because of the group sessions, I am no longer afraid to speak to the nurse. I ask questions when I don't understand, and this has helped me follow the advice given during my clinic visits."*(Pregnant woman, Health Centre)

### Perception and experience of healthcare providers on G-ANC

Healthcare providers who implemented and facilitated G-ANC reported their positive experiences, highlighting its effectiveness in improving maternal health outcomes, improving patient engagement, and fostering a supportive environment for both providers and pregnant women. HCPs acknowledged that G-ANC allowed them to deliver health education more efficiently, reaching multiple women at once instead of repeating the same information individually. One midwife shared,

*"With G-ANC, we can educate many women at the same time, ensuring they receive consistent and accurate information. It saves time and improves understanding." Health Care Provider,* District Hospital 2

Another key benefit recognized by HCPs was the improved rapport and trust between providers and pregnant women. In conventional ANC, women often hesitate to ask questions due to time constraints or fear of being judged. Improved communication helped providers address misconceptions and ensured women were better prepared for childbirth. However, G-ANC created an open and interactive space where women felt comfortable discussing their concerns. A nurse explained,

*"In one-on-one visits, many women remain silent, but in a group setting, they are more open to asking questions and sharing their concerns. It makes our work easier and more impactful."* Health Care Provider, Dispensary 2

HCWs also observed that G-ANC strengthened peer support among pregnant women and promoted positive behavioral changes. Women, inspired by their peers, became more proactive in adhering to recommended antenatal care practices, such as taking iron supplements, Intermittent Preventive Treatment in pregnancy (IPTp) uptake, and preparing for institutional delivery. One health provider noted,

*"Women who were initially reluctant to follow ANC recommendations became more motivated after seeing others share their positive experiences. The group setting created a sense of accountability and encouragement." Healthcare Provider,* District Hospital 2

*"The effect on maternal health is evident. Women are more informed, confident, and engaged. We need to invest more in scaling up this intervention," Facility In -charge* Health Centre 1

### Health managers' perception and experience with G-ANC

Health managers overseeing the implementation of services in Chato and Geita districts expressed strong support for the ANC approach, recognizing its transformative effect on maternal health care services. They highlighted improvements in service delivery, increased ANC attendance, and better engagement between healthcare providers and pregnant women. Some managers noted that G-ANC has led to more efficient use of resources and improved health outcomes. One district SRH coordinator explained,

*"G-ANC helped our facilities to increase ANC attendance indicator. Women enjoy coming to the sessions because they receive care and learn from each other in a supportive environment."* Health Coordinator District 1

They acknowledged that unlike the conventional ANC, which often resulted in brief interactions between providers and clients, G-ANC enhanced a more interactive and trusting environment.,

*"At Nzera hospital we have seen a shift in how women perceive ANC services. They are more engaged, ask more questions, and actively participate in discussions, which was not the case before...This change was attributed to the group-based approach, which encouraged collective learning and reduced fear or hesitation among pregnant women". Health Coordinator, District 2*

Health managers also appreciated the efficiency of G-ANC in optimizing healthcare workers' time and reducing workload pressures. Instead of repeating the same health education topics individually, providers could deliver key messages to multiple women simultaneously. They also strongly supported scaling up G-ANC, recognizing its potential to improve maternal health outcomes.

*"With limited staff, G-ANC helps us reach more women effectively. Providers can dedicate more time to meaningful discussions rather than rushing through individual consultations. The effect is clear…women are more informed, confident,*

*and proactive about their health. We need to invest in infrastructure and training to sustain and expand this approach,"*
Health Coordinator, District 2

## Experience on practicality of Group Antenatal Care

G-ANC required careful planning to ensure that health facilities had the necessary resources to run group sessions effectively. Key resources included trained healthcare workers, sufficient space for group meetings, and the provision of essential materials such as educational tools (maternal booklets) and proper health records. Healthcare workers reported that organizing and facilitating G-ANC sessions required more initial time and effort compared to individual visits. One midwife noted,

*"At first, it was challenging to organize the group sessions because we had to coordinate schedules in the G-ANC Cohort tracker or ANC registers and ensure that we had enough space. But now that we have the process in place, we are trying our best." Healthcare Provider, Dispensary 1*

Training healthcare workers to facilitate G-ANC was important to its success. HCWs noted the importance of equipping them with the skills needed to manage group dynamics and address the diverse needs of pregnant women in a group setting. A nurse explained,

*"The training we received in Katoro made a huge difference in how we conduct the sessions. It's not just about providing information…it's about managing the group and facilitating discussion." Healthcare Provider,* Health Centre 2

Integrating G-ANC into routine maternal health services was seen as a practical and beneficial approach to improve service delivery efficiency. It allowed healthcare workers to educate multiple women simultaneously while still providing individualized care and counselling when necessary. This was particularly important in areas with high patient volumes, where conventional one-on-one consultations could lead to long waiting times and overcrowding. One facility in-charge said,

*"G-ANC allows us to manage our time better. It reduces waiting times for women and ensures that we can reach more women with the same resources, especially in facilities with many clients like Bwanga and Katoro I think..." Facility In-charge,* Dispensary 2

The integration of G-ANC with other maternal, child health, and community services was a key feature of its implementation. Through leveraging existing health systems and services, G-ANC promoted a holistic approach to care. The integration was achieved through strategic collaboration between healthcare workers from different units such as labour ward and the laboratory. Another area of integration is family planning services. During G-ANC sessions, healthcare workers introduced a full session discussion about postpartum family planning, which allowed women to make informed decisions about their reproductive health before delivery. A healthcare worker explained,

*"We ensure that women understand their options for family planning after childbirth. We encourage them to choose the methods before birth, and this is a full session discussion".* Healthcare Provider, District Hospital 1

Women, in turn, appreciated the opportunity to receive detailed family planning counselling during their group care sessions. Unlike conventional ANC women receive partial counselling of family planning issues. A participant shared,

*"It's helpful to learn about family planning while I am still pregnant. I feel more prepared for after delivery." Pregnant woman, Dispensary 1*

## Acceptability of Group Antenatal Care among women

For women, G-ANC was accepted as a valuable alternative to conventional antenatal care. Many women appreciated the opportunity to engage with other pregnant women, which fostered a sense of community and emotional support. A pregnant woman mentioned

*"I feel less alone now because we talk about our pregnancies together, and I can learn from others' experiences. It makes me more confident about my pregnancy."* Pregnant woman, Dispensary 2

The shared learning environment allowed women to feel more empowered in managing their pregnancies, and the group format was often described as both informative and reassuring. One woman commented,

*"I used to feel anxious going to individual appointments, but in a group, we share our worries and talk openly. It feels like we're all in it together."* Postnatal woman, Health Centre 1

While most women were positive about G-ANC, there were some who expressed concerns about group dynamics. One participant mentioned feeling shy or uncomfortable discussing personal health issues in front of others, especially in larger groups. However, the majority of women acknowledged that the benefits of being in a group far outweighed these concerns. This participant noted,

*"Sometimes, I feel uncomfortable asking certain questions in front of so many people. It would be better if we had smaller groups."* Postnatal woman, Health Centre 2

## Acceptability of G-ANC among healthcare providers

Generally, healthcare providers who were involved in this study perceived G-ANC as an appropriate approach for delivering antenatal care services. They recognized the advantages of the group care in terms of time management and increased efficiency. Also, they appreciated the opportunity to provide more interactive, participatory sessions. A midwife expressed,

*"G-ANC has allowed us to engage with many women at once, which helps reduce the pressure on our time. It's a more organized way to deliver care, women feel more involved in their care, and it's much easier to address common questions in a group than one by one."* Healthcare Provider, District Hospital 2

*"At first, we were unsure about how to manage the group dynamics and ensure every woman got the individual attention they needed. With more experience, we've found ways to balance the group discussions with one-on-one support."* District Hospital 1

## Acceptability of G-ANC among the health managers

District health managers were generally supportive of G-ANC, as they experienced the G-ANC approach as a means to promote service delivery coverage and improve pregnant outcomes. A district health coordinator noted,

*"G-ANC has increased the number of women attending antenatal care regularly which is a big win for our Geita region. It also reduces the workload on individual healthcare appointments."* Health Coordinator, District 1

It was recognized that G-ANC could potentially reduce maternal and neonatal health risks by increasing the number of women attending regular check-ups and receiving timely care. However, one health manager expressed concerns about the initial costs of implementing G-ANC, particularly after the Mlinde Mama project closeout.

*"While we support the idea, the costs involved in setting up the program is a bit challenging especially if PHIT leave the project. But we saw the value it brought in terms of both health outcomes and service efficiency, but we don't have money to print maternal booklets and to replace rechargeable batteries for BP machines."* Health Coordinator, District 2

## Barriers to successful implementation of G-ANC

Several barriers were identified that could limit effective implementation of G-ANC and its acceptability. Privacy and confidentiality concerns were frequently raised by women, particularly regarding the sharing of sensitive personal information in a group setting. Some women expressed fear that details disclosed during sessions could be discussed outside the group, leading to social discomfort or stigma within their communities.

*"When you are in a group, you are not sure who will keep your secrets. Some issues are very personal, like problems in the pregnancy or things happening at home, and if they spread, it can affect your relationship with your husband or even how neighbours treat you. That fear sometimes makes you keep quiet, even when you need help."* Pregnant woman, Dispensary 1

Others noted that this concern made them hesitant to fully participate or ask questions related to complications or personal challenges:

*"There are things you want to ask the nurse, but you keep quiet because you don't want everyone to know your problem, especially when it is something very personal. "* Postnatal woman, Health Center 2

Economic barriers also influenced participation in G-ANC. Women reported that lack of money for transport, food, or childcare reduced their ability to attend sessions consistently. One postnatal woman noted:

*"Even if the service is good, you still need money to reach there. Sometimes I had to choose between attending the group and buying food for my children."* (Postnatal woman, District Hospital 2)

In addition, male partner support played a role in enabling or constraining participation. Providers and managers also acknowledged that limited male engagement could negatively affect women's attendance and continuity in care. Several women described situations where partners were unsupportive or questioned the value of group-based care, limiting women's autonomy to attend sessions.

*"My husband asked why I should go many times for the same check-up. When he refuses to support me, I miss some meetings."* (Pregnant woman, Health Center 1)

*"Sometimes my partner would say the group meetings are a waste of time and that one visit is enough. For me, when my husband was not involved or informed, I missed some sessions because I did not want to cause conflict at home."* Postnatal woman, District Hospital 2

Geographical distance and transport challenges were particularly pronounced in two rural facilities. Women living far from facilities or community meeting points reported difficulty attending sessions, especially in later stages of pregnancy or during the rainy season.

*"The road is bad, and when it rains, it is impossible to reach the health centre. That made me miss some group meetings."* Pregnant woman, Dispensary 1

*"During the rainy season, even if you want to attend the group, it becomes very difficult. The paths are flooded, boda boda (motorcycles) refuse to come,… how can you attend the clinic".* Postnatal woman, Dispensary 2

From the healthcare providers' perspective, organizational challenges within facilities were highlighted. Providers emphasized the importance of protected time for G-ANC sessions and noted that frequent interruptions undermined the quality of group discussions.

*"If we are called to attend emergencies or other clinics during the group session, we cannot concentrate. Women notice this and feel the session is not taken seriously."* Healthcare provider, District Hospital 2

Health managers echoed this concern, noting that staff shortages and competing service demands made it difficult to consistently allocate uninterrupted time for group care.

*"We support G-ANC, but with few staff, it is not always easy to free one provider for the whole session without interruptions."* Health Coordinator, District 1

## Discussion

This study explored the perspectives of pregnant or recently delivered women, healthcare providers, and health system managers to explore experience and acceptability G-ANC implementation. The present study demonstrates that women are satisfied with G-ANC. Respondents highlighted positive experience including some befts of G-ANC such as enhanced health care knowledge, emotional well-being during pregnancy, and enhances provider-client interaction in healthcare delivery. These findings are consistent with previous studies in sub-Saharan Africa and other low-resource settings that have demonstrated the benefits of G-ANC in improving maternal health care. [5,12,15,26]. The shared positive experiences reported by women in this study align with findings from Ghana and Kenya, where G-ANC fostered social cohesion, peer support, and improved health literacy [5,15]. This may be attributed by interactive learning environment and emotional support fostered through shared experiences.

Moreover, the enhancement of communication and trust between pregnant women and providers observed in this study corroborates evidence from the studies conducted in Ghana and Burkina, where G-ANC increased client-provider interaction and improved understanding of maternal issues [5,27]. This reinforces the notion that G-ANC, through its participatory design, facilitates more meaningful engagement compared to conventional one-on-one care. The reasons for high satisfaction have been reported in other studies, these include self-measuring of blood pressure or body weight and peer to peer learning [5,28,29].

However, our study also highlights a unique finding in which women reported gaining confidence in communicating health messages or advocating for maternal health within their communities after community-based meetings. These unique findings may be due to fact that some G-ANC meetings were conducted in the community. This community-based ripple effect has not been extensively documented in previous studies and warrants the need of further exploration. However, previous studies have demonstrated how the community-based health services promote satisfaction and uptake of ANC services compared to facility-based ANC services [30–32]. Community-based delivery of ANC sessions expands women's exposure to health information within familiar and socially supportive environments. This reduces social and physical barriers to participation, increased attendance and reinforce their confidence [30,31].

Health managers and HCPs this study recognized the contribution of G-ANC in increasing ANC attendance, improving quality of care, and reducing workload. The findings on improved perceived quality of care and ANC attendance collaborates with the similar studies that were conducted in Burkina Faso [27], Rwanda [17] and Malawi [9]. Reasons for reduced workload reported in this study include group-based health education that replaces repetitive long individual counselling.

In conventional ANC, providers often deliver the same information repeatedly to individual clients (e.g., nutrition, danger signs, birth preparedness) [12,33]. G-ANC allows providers to deliver standardized, stage-appropriate health education to multiple women simultaneously, substantially reducing repetition and consultation time per client [5,12,13,27]. In addition, improved appointment organization and cohort scheduling enhance workflow efficiency. G-ANC uses cohort-based scheduling, which reduces unpredictable client flow, minimizes congestion, and allows providers to plan their time more efficiently compared to walk-in, one-on-one ANC approach [13,23,27]. This finding is important for practice because it demonstrates that G-ANC can improve service efficiency and provider workload management without compromising quality of care. This is critical in health systems facing chronic human resource constraints like Tanzania.

On the other hand, our findings that G-ANC contributed to reduced workload for healthcare providers are inconsistent from evidence in other previous studies. For instance, a systematic review involving nineteen papers from nine countries demonstrated that G-ANC increase workload to HCPs [34]. Another study has shown that G-ANC does not consistently reduce provider workload, particularly during early phases of implementation or in settings where adequate training, staffing and system-level support are insufficient [35]. Providers invest additional effort in organizing cohorts, scheduling group sessions, organizing educational materials and managing group dynamics. In understaffed facilities, these added responsibilities may increase rather than reduce perceived workload, especially when G-ANC is implemented alongside conventional ANC [34,35]. A common practice in pilot studies, G-ANC is layered onto existing ANC services without reducing individual visits or adjusting staffing norms. Providers therefore deliver both group sessions and routine ANC, effectively duplicating work rather than streamlining it [13,17,23]. The highlighted findings underscore the need for context-sensitive implementation, rather than assuming uniform workload benefits across settings.

Participants in this study demonstrated high acceptability of G-ANC. In the present study, acceptability was not treated as a single construct; rather, it was interpreted in reference to definitions from the existing literature. Sekhon et al. define acceptability as the extent to which an intervention is perceived as appropriate, satisfactory and aligns with stakeholders' expectations and values. Similar considerations were applied when reporting acceptability in qualitative studies, as demonstrated in a study assessing the acceptability and impact of G-ANC among women in Burkina Faso [27].

Acceptability of GANC in this study was supported by integration of intervention with other maternal health services like postpartum family planning and community health support systems. Group care promotes repeated health messages, support follow-up, and facilitated referrals at the community level for G-ANC clients [12,5,20,27]. In this study women and HCPs demonstrated preference of G-ANC compared to conventional ANC, appreciating the more interactive format that facilitated open dialogue and strengthened provider–client relationships. For example, a women noted "*Because of the group sessions, I am no longer afraid to speak to the nurse. I ask questions when I don't understand, and this has helped me follow the advice given during my clinic visits* ". A previous study in Ghana reported similar findings where women and HCPs endorsed G-ANC due to its high interactive approach [5].

Another construct of acceptability demonstrated in this study was the extent to which G-ANC aligned with women's values and social norms. Women perceived the group care as consistent with communal learning, mutual support, and shared problem-solving. One woman noted "*It felt reassuring to be in a group with other pregnant women who were going through the same journey as me…We encouraged each other and became like a family.*" This alignment with existing values enhanced comfort, trust, and willingness to participate in G-ANC, thereby reinforcing its acceptability and sustained engagement. In Geita, where social and cultural traditions strongly shape women's lives, the introduction of G-ANC encountered minimal resistance from pregnant women. These finding echoes community engagement theory, which posits that health interventions are more likely to be accepted and sustained when they align with existing community values, social norms, and collective practices [36]. This aligns with global strategies and recommendations for comprehensive, integrated people-centred care health systems [37]. The study adds to the growing evidence that G-ANC is not merely a service delivery innovation but a transformative approach that fosters peer-to-peer learning and empowerment of women for self-care.

On the other side, while the majority of participants endorsed G-ANC, some women expressed concerns about discussing personal health issues in group settings. This hesitation reflects cultural norms regarding privacy. This is consistent with findings from other studies, where women preferred individual consultations for sensitive topics [5,15,27]. However, unlike in some other settings where such discomfort significantly affected participation [27], this study found that most women overcame initial shyness and eventually valued the shared experience.

Several challenges to successful implementation of G-ANC were noted. This study did not highlight substantial male involvement in antenatal care through G-ANC. This may be explained by the way male engagement was structured during implementation, whereby men were primarily involved in specific, targeted sessions such as the third meeting focused on family planning. Women showed how this affected their participation at some point. Eg. A woman said ""*Sometimes my partner would say the group meetings are a waste of time and that one visit is enough. For me, when my husband was not involved or informed, I missed some sessions because I did not want to cause conflict at home"*. This is contrary to the previous evidence from Malawi showing that group care encouraged male participation in maternal health services [9].

The absence of this theme could suggest contextual or cultural differences in partner involvement in maternal health in Geita or a limitation in how group sessions were structured. Frequent interruptions due to competing clinical duties or emergency cases were reported to undermine the quality and flow of group sessions, sometimes leading to rushed discussions and reduced participant engagement. Similar challenges have been documented in previous studies from low-resource settings, where inadequate staffing and competing service demands limited providers' ability to consistently deliver high-quality group care [17,27,35]. Evidence from South-West Netherlands has shown that without deliberate scheduling and planning, G-ANC sessions are vulnerable to disruption, which can negatively affect both provider experience and women's satisfaction [35]. These findings underscore the importance of health system readiness, including staffing norms and protected clinic time, to ensure that G-ANC can be delivered as intended and sustain its perceived benefits.

While this study generated valuable insights, it has several limitations. First, the data were drawn from a specific rural geographical context (six facilities), which may limit generalizability to other regions with different sociocultural dynamics or health system capacities. Future studies could expand to include a more diverse sample across multiple settings. In addition, the analysis did not assess economic feasibility or cost-effectiveness, which are important considerations for informing national-level decision-making. Although the overall intervention was a mixed study, this article did incorporate quantitative components, such as cost–benefit or economic analysis, which limit a more comprehensive understanding of the intervention's potential in terms of costs effectiveness.

A key strength of this study is its comprehensive assessment of acceptability and experience, which extended beyond behavioural indicators such as attendance or continued participation. While engagement and adherence are commonly used proxies for acceptability of interventions, they do not capture the underlying cognitive and emotional factors that shape individuals' decisions to participate in or disengage from an intervention. By incorporating direct participant-reported accounts through in-depth interviews and FGDs, this study was able to explore experiences, perceptions, and attitudes toward G-ANC, providing a richer and more nuanced understanding of acceptability than behavioural measures alone.

## Conclusion

Findings revealed high satisfaction and acceptability of G-ANC among women and HCPs. Women reported increased knowledge on maternal health, emotional support, and enhanced communication with providers. Healthcare providers noted improved efficiency, better patient engagement, and peer support among clients. Health managers demonstrated the contribution of G-ANC in promoting ANC attendance and more interactive care. However, challenges such as limited space, need for facilitator training, and resource constraints were highlighted by HCPs and Health Managers. Some women experienced shyness or hesitancy to share personal issues during the G-ANC sessions. Overall G-ANC was positively experienced and is acceptable intervention in the Tanzanian context, with significant benefits for peer support and service delivery efficiency. These findings add evidence to other existing literature supporting the adoption of this

intervention in low resource settings. However, succeful implementation requires investments in infrastructure (space for G-ANC meetings), adequate and trained HCPs, service integration and male engagement. Institutionalizing G-ANC as a standard antenatal care could be the best option to address the low coverage of ANC visits, suboptimal quality care, and promotion of institutional deliveries.

## Supporting information

**S1 Table. Consolidated criteria for reporting qualitative studies (COREQ) checklist.**
(DOCX)

---

### Key messages

**What is already known on this topic:**

- Group Antenatal Care has been implemented in various settings globally, showing improvements in maternal health literacy, peer support, and satisfaction with care.

- There is limited research from other low-resource settings. However, available evidence suggests that G-ANC can improve antenatal care attendance and enhance provider-client interaction.

- Despite growing global interest, limited evidence exists on the acceptability and feasibility of G-ANC in routine health systems in Tanzania.

**What this study adds:**

- Women, healthcare providers, and health managers reported largely positive experiences with G-ANC, highlighting improved maternal health knowledge, peer support, and more open communication between clients and providers.

- Group care was perceived as acceptable by women, healthcare providers, and health managers, driven by its alignment with community values, interactive group format, and integration with existing maternal health and community health systems.

**How this study might affect research, practice, or policy**

- This study contributes context-specific qualitative evidence on how G-ANC is experienced and accepted within routine public-sector health facilities in a non-urban Tanzanian setting. The findings highlight the importance of examining experiential and acceptability dimensions and point to the need for future mixed-methods and economic evaluations to assess cost, sustainability, and comparative effectiveness across diverse contexts.

- For practice, the study offers insights into how G-ANC can be delivered within existing health system constraints. These findings can inform implementers seeking to adapt G-ANC to similar low-resource settings, emphasizing the need for adequate training, space, and alignment with routine workflows.

- For policy, this study does not provide evidence for immediate national scale-up, but it does offer implementation-relevant insights that can support informed decision-making. Understanding stakeholder experiences and acceptability may help policymakers assess whether and how G-ANC could be incorporated into antenatal care strategies, piloted in additional regions, or refined within existing maternal and newborn health programs. Importantly, the findings underscore that successful adoption of G-ANC requires attention to local context, health system capacity, and community values rather than a one-size-fits-all approach.

## Acknowledgments

We extend our heartfelt appreciation to the healthcare providers, facility in-charges, and health managers at the participating health facilities in Geita Region for their invaluable time, cooperation, and commitment to this research. Special thanks go to the Geita Regional Health Management Team (RHMT) and the Council Health Management Teams (CHMTs) of Chato and Geita DC for their technical and logistical support throughout the study period. We acknowledge the support of the Ministry of Health (MoH) and the President Office - Regional Administration and Local Government (PORALG) for their policy guidance and facilitation of approvals essential for data collection and implementation of the G-ANC in the public facilities. The corresponding author (AK) is supported by the Consortium for Advanced Research Training in Africa (CARTA) as a PhD Fellow. CARTA is jointly led by the African Population and Health Research Center and the University of the Witwatersrand and funded by the Carnegie Corporation of New York (Grant No. G-CS-24-62008 and G-PS-23-60922), Sida (Grant No: 16604), Norwegian Agency for Development Cooperation (Norad) (Grant No: QZA-21/0162), Oak Foundation (Grant No. OFIL-24-091) and the Science for Africa Foundation to the Developing Excellence in Leadership, Training and Science in Africa (DELTAS Africa) programme (Del-22-006) with support from Wellcome and the UK Foreign, Commonwealth & Development Office and is part of the EDCPT2 programme supported by the European Union. The statements made and views expressed are solely the responsibility of the Author.

## Author contributions

**Conceptualization:** Alen Kinyina, Augustino Hellar, Raymond Bandio, Phineas Sospeter, Yusuph Kulindwa, Ntuli Kapologwe, Frank Phiri, Wilfred Kafuku, Cyprian Mtani, Husna Athumani.

**Data curation:** Alen Kinyina, Augustino Hellar, Frank Phiri.

**Formal analysis:** Alen Kinyina, Augustino Hellar, Frank Phiri.

**Funding acquisition:** Augustino Hellar, Raymond Bandio, Hamid Mandali, Yusuph Kulindwa, Frank Phiri, Wilfred Kafuku, Husna Athumani.

**Investigation:** Alen Kinyina.

**Methodology:** Alen Kinyina, Augustino Hellar, Raymond Bandio, Hamid Mandali, Phineas Sospeter, Yusuph Kulindwa, Frank Phiri, Isaac Lyatuu, Wilfred Kafuku, Cyprian Mtani.

**Project administration:** Alen Kinyina, Augustino Hellar, Raymond Bandio, Hamid Mandali, Ahmad Makuwani, Abubakari Munga, Phineas Sospeter, Yusuph Kulindwa, Ntuli Kapologwe, Frank Phiri, Isaac Lyatuu, Wilfred Kafuku, Cyprian Mtani, Husna Athumani, Omari Sukari.

**Resources:** Alen Kinyina, Augustino Hellar, Raymond Bandio, Yusuph Kulindwa, Wilfred Kafuku, Cyprian Mtani, Husna Athumani.

**Supervision:** Alen Kinyina, Augustino Hellar, Raymond Bandio, Hamid Mandali, Abubakari Munga, Yusuph Kulindwa, Ntuli Kapologwe, Frank Phiri, Isaac Lyatuu, Wilfred Kafuku, Cyprian Mtani, Husna Athumani.

**Validation:** Augustino Hellar, Hamid Mandali, Ahmad Makuwani, Phineas Sospeter, Yusuph Kulindwa, Ntuli Kapologwe, Isaac Lyatuu, Cyprian Mtani, Omari Sukari, James Tumaini Kengia.

**Writing – original draft:** Alen Kinyina, Augustino Hellar, Raymond Bandio, Hamid Mandali, Ahmad Makuwani, Abubakari Munga, Frank Phiri, Isaac Lyatuu, James Hellar, James Tumaini Kengia.

**Writing – review & editing:** Alen Kinyina, Augustino Hellar, Raymond Bandio, Hamid Mandali, Ahmad Makuwani, Abubakari Munga, Phineas Sospeter, Yusuph Kulindwa, Ntuli Kapologwe, Frank Phiri, Isaac Lyatuu, Wilfred Kafuku, Cyprian Mtani, James Hellar, Omari Sukari, James Tumaini Kengia, Erin Jones-Avni, Hermes Gichane.

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
