## [Decision Letter · Decision Letter 0]

18 Sep 2025

Dear Dr. Kinyina,

We look forward to receiving your revised manuscript.

Kind regards,

Hannah Tappis, DrPH, MPH

Academic Editor

PLOS ONE

Journal Requirements:

https://journals.plos.org/plosone/s/file?id=wjVg/PLOSOne_formatting_sample_main_body.pdf  and  and  and  and

2. In the ethics statement in the Methods, you have specified that verbal consent was obtained. Please provide additional details regarding how this consent was documented and witnessed, and state whether this was approved by the IRB.

Reviewers' comments:

Reviewer's Responses to Questions

**Comments to the Author**

1. Is the manuscript technically sound, and do the data support the conclusions?

Reviewer #1: Yes

Reviewer #2: No

Reviewer #3: Yes

2. Has the statistical analysis been performed appropriately and rigorously?

Reviewer #1: N/A

Reviewer #2: Yes

Reviewer #3: N/A

3. Have the authors made all data underlying the findings in their manuscript fully available?

Reviewer #1: Yes

Reviewer #2: Yes

Reviewer #3: No

4. Is the manuscript presented in an intelligible fashion and written in standard English?

Reviewer #1: Yes

Reviewer #2: Yes

Reviewer #3: Yes

Reviewer #1: I am very grateful for the opportunity to review this manuscript, below are some suggestions with the intention of contributing to the development of the text.

Title: In the title I believe it is more appropriate not to use acronyms or abbreviations.

Abstract: I miss listing the categories created to present the results and I believe it needs to be adapted after making updates to the text.

Background:

Some paragraphs are very long, reaching 16 lines, I suggest reviewing and distributing the information better. I also suggest not starting with acronyms or abbreviations.

I believe it is better not to include data relating to unpublished results of studies, as those that are already published allow for a good contextualization of the problem and justify studying the G-ANC model.

Aim: clearly described at the end of the introduction.

Methods: I believe the themes covered in the interviews and focus groups could be included in the text, or the triggering question, for example. Another important point is that focus groups are usually conducted with a facilitator, a co-facilitator, and an observer. I'd like the authors to reflect on the different configurations used.

I'd also like to better understand how the data analysis was conducted—whether by one researcher or two researchers?

Similarly, I'd like to know how the internal and external validation of the research was ensured.

Were any verification instruments used, such as the COREQ checklist?

I believe it is important to describe how the sample was closed and how this process was within the sample triangulation used.

I would also like a description of the research questions, as the researchers state that "Themes were identified using a combination of prior issues included in the discussion and interview guides that were informed by the research questions as well as emergent issues raised by the participants," however, these questions are not presented in the text.

I believe that this section should only contain the results of the study: description of the categories and vignettes that illustrate them. I see no need to cite references about the method; it is sufficient to explain it in the methods section.

Discussion: there are very long sections, I suggest distributing the information better

Conclusion: I believe that measuring satisfaction levels and stating that high levels were identified is not possible with qualitative research. I suggest changing the verbs used to reflect what the study actually shows.

I also suggest that the authors rewrite the section, reflecting on the challenge of assuming in a single study that the model is the best one to fill this gap.

In addition, I suggest that the How this study might affect research, practice, or policy sections be rewritten in a more realistic way and that the authors provide a table/chart that characterizes the study participants and a figure that shows the process of constructing the categories.

Success to the authors.

Reviewer #2: The qualitative design of this study is appropriate for the stated objectives, the methods (FGDs, KIIs, thematic analysis using NVivo, guided by the Bowen feasibility framework) are well aligned to the research questions, and the findings are clearly supported by the data presented. Inter-coder reliability checks were not included in the summary, so this would be one noted weakness.

Reviewer #3: This is an excellent addition to the G-ANC literature and should be valuable evidence to support policy changes in Tanzania. Overall the paper is well structured and well-written. It in some ways feels like 2 papers: 1 on experiences, and another on feasibility. It's good to keep them combined, but suggesting some re-structuring as well as other feedback:

1) It's helpful when research articles clearly state the study objectives being presented in the paper. In this case the results section covers 3 topics (e.g., experience, acceptability, feasibility) and each can be framed more explicitly as an objective. This would reshape the methods by adding a definition of acceptability, in addition to the feasibility definition and reference. The Bowen framework includes acceptability as a dimension of feasibility, but other Implementation Research frameworks (e.g., Proctor) define them separately. It would then follow the results section structure, and help restructure the Discussion around the 3 objectives. If the authors don't have enough additional content to add, then would suggest there be 2 explicit study objective on experience and on feasibility (with acceptability as one of the themes).

2) The paper can benefit from more G-ANC research context into the Background and Discussion so it's building on the full range of G-ANC in LMIC literature on the 2 outcomes (acceptability and feasibility), as well as the previous trial in Tanzania. There are many other studies that have reported on these 2 domains that can bring more context to these findings, and enhance the discussion by comparing what is similar or different from the Geita implementation. There are suggested citations as well as comments on the listed references for consideration. The background and Discussion could be revised to focus on those 2 outcomes, in addition to what's presented on experiences.

3) The description of participants throughout the paper is not always consistent which can be confusing, so this needs to be reviewed and streamlined from the abstract through the paper. Pregnant women and those who recently delivered were both FGD participants, but described differently in different sections. Also the description of providers (which cadres, and including CHWs who aren't typically called providers) and health managers can be explained in detail in the Methods section, and then terminology made consistent throughout.

4) Methods: Consider describing more how the sampling was done, e.g., how the 1 ANC provider per facility was selected. Why more recently delivered mothers were invited than pregnant women, and if the FGDs were pulled from each facility or across facilities into a mixed group.

5) Results: The results section is very rich with a good selection of quotes. Suggest that there be greater anonymity in the descriptions before and tagged to the quotes, which seem likely very easy to identify the individual. The author team can consider dropping the facility name and using just the facility type, as well as considering not specifying the participants' cadre or position in the text.

6) Results: The experiences section is long and has similar findings to other G-ANC implementation. It could be shortened if the feasibility section can be expanded. Would make the acceptability the second sub-section, then end with feasibility. If the qualitative data are available, it is suggested to add more detail specific to the feasibility of G-ANC implementation on the most practical issues and elaborate on any barriers and challenges. Also the methods describe alot about the facility types, but the results are not highlighting much on different experiences at the different types of facilities. This would be valuable to understand if there were different implementation challenges at dispensaries vs. hospitals. For example, staffing shortages are a challenge in many LMICs and CHWs' roles could be explored more here. It would also be helpful to understand more from the health managers/system perspective on the practical issues around the feasibility of implementation, with less focus on the effectiveness (retention in ANC). Hopefully there is more qualitative content to pull into the results to enrich the feasibility content.

7) Discussion: As noted in the first point, there could be more in the discussion about how these findings on acceptability and feasibility compare to other studies.

There are more comments in the PDF for consideration.

.

Reviewer #1: No

Reviewer #2: No

Reviewer #3: **Yes:** Stephanie SuhowatskyStephanie SuhowatskyStephanie SuhowatskyStephanie Suhowatsky

---

## [Author Response · Author response to Decision Letter 1]

4 Feb 2026

GENERAL RESPONSE FROM THE AUTHORS:

We sincerely appreciate the constructive comments provided by the reviewers and the editorial team. We have addressed all comments comprehensively, including every issue highlighted in the annotated PDF. We believe that the manuscript is now well structured, coherent, and of sufficient quality to meaningfully contribute to the evidence base on the implementation of group antenatal care in low-resource settings.

Comments from Editorial Team

2. In the ethics statement in the Methods, you have specified that verbal consent was obtained. Please provide additional details regarding how this consent was documented and witnessed, and state whether this was approved by the IRB.

Response: Thank you for this important clarification request. We confirm that the use of verbal consent was reviewed and approved by the Institutional Review Board.

Response: There is no restriction on access to the de-identified dataset generated by this study. For the qualitative component, no personally identifiable information was collected, and interviews were either de-identified at the point of transcription. Contact information for requesting access to the data has been provided.

b) If there are no restrictions, please upload the minimal anonymized data set necessary to replicate your study findings to a stable, public repository and provide us with the relevant URLs, DOIs, or accession numbers.

Response: A link has been provided: https://doi.org/10.7910/DVN/RLAJMC

Response: Thank you, all citations and references have reviewed.

Response: Thank you, all references have reviewed.

REVIEWER 1

Reviewer #1: I am very grateful for the opportunity to review this manuscript, below are some suggestions with the intention of contributing to the development of the text.

Title: In the title I believe it is more appropriate not to use acronyms or abbreviations.

Response: This is noted and well addressed.

Abstract: I miss listing the categories created to present the results and I believe it needs to be adapted after making updates to the text.

Response: This is noted and addressed

Background:

Some paragraphs are very long, reaching 16 lines, I suggest reviewing and distributing the information better. I also suggest not starting with acronyms or abbreviations.

Response: This is noted and well addressed.

I believe it is better not to include data relating to unpublished results of studies, as those that are already published allow for a good contextualization of the problem and justify studying the G-ANC model.

Response: We agree with the reviewer’s observation that referencing unpublished study results may limit transparency and verifiability. In response, we have revised the manuscript to remove references to unpublished findings and now rely exclusively on peer-reviewed and publicly available evidence.

Aim: clearly described at the end of the introduction.

Response: Thank you.

Methods: I believe the themes covered in the interviews and focus groups could be included in the text, or the triggering question, for example. Another important point is that focus groups are usually conducted with a facilitator, a co-facilitator, and an observer. I'd like the authors to reflect on the different configurations used.

Response: Thank you for this comment. Additional details have been added, and the covered themes have now been included.

I'd also like to better understand how the data analysis was conducted—whether by one researcher or two researchers?

Response: Thank you for this comment. The data analysis was conducted by two researchers among the research team, who independently analyzed the data and then compared their findings to ensure consistency. Any discrepancies were discussed and resolved through consensus. More details have been added in the manuscript.

Similarly, I'd like to know how the internal and external validation of the research was ensured.

Were any verification instruments used, such as the COREQ checklist?

Response: Thank you for this important methodological question.

Internal validation was ensured through multiple strategies. First, methodological triangulation was employed by collecting data from three distinct stakeholder groups (women, healthcare providers, and health managers) using both FGDs and KIIs, allowing for cross-verification of findings across sources. Second, interviewer training and use of standardized, semi-structured guides enhanced consistency in data collection. Third, all interviews were audio-recorded, transcribed verbatim, and translated from Kiswahili to English, with accuracy checks conducted by bilingual members of the research team. Fourth, team-based coding and iterative theme refinement were conducted and regular analytic discussions to resolve discrepancies and enhance interpretive rigor. Data collection continued until thematic saturation was achieved. The consolidated criteria for reporting qualitative research (COREQ) checklist was used to enhance the rigour and transparency of the study’s reporting.

More details have been added in the manuscript.

I believe it is important to describe how the sample was closed and how this process was within the sample triangulation used.

Response: Thank you for this comment, some details have been addressed. Sampling and sample closure were guided by the principles of purposive sampling, triangulation, and thematic saturation, rather than by a predetermined sample size. Participants were selected to ensure representation of key stakeholder groups involved in the design, delivery, and receipt of G-ANC, namely women receiving care, healthcare providers, and health system managers.

Sample closure was reached when thematic saturation was achieved, defined as the point at which additional FGDs or KIIs no longer yielded new concepts, or substantive insights relevant to the study objectives. Saturation was assessed separately within each participant group and then across groups.

Sample triangulation was achieved through the inclusion of multiple perspectives across different roles (service users, providers, and managers), facility levels (hospital, health centre, and dispensary), and geographical contexts (urban, peri-urban, and rural settings). This triangulated sampling strategy allowed for comparison and validation of findings across participant categories and service delivery contexts.

I would also like a description of the research questions, as the researchers state that "Themes were identified using a combination of prior issues included in the discussion and interview guides that were informed by the research questions as well as emergent issues raised by the participants," however, these questions are not presented in the text.

Response: Thank you for highlighting this important point. In response, the core thematic areas explored in the interviews and focus group discussions have been described.

I believe that this section should only contain the results of the study: description of the categories and vignettes that illustrate them. I see no need to cite references about the method; it is sufficient to explain it in the methods section.

Response: Thank you, this section has been revised.

Discussion: there are very long sections, I suggest distributing the information better

Response: Thank you for this helpful suggestion. We agree that some sections of the Discussion were overly long. In response, we have revised and reorganized the Discussion section, redistributing content to improve flow, readability, and logical progression of arguments.

Conclusion: I believe that measuring satisfaction levels and stating that high levels were identified is not possible with qualitative research. I suggest changing the verbs used to reflect what the study actually shows.

Response: Thank you the conclusion section has been revised.

I also suggest that the authors rewrite the section, reflecting on the challenge of assuming in a single study that the model is the best one to fill this gap.

In addition, I suggest that the How this study might affect research, practice, or policy sections be rewritten in a more realistic way and that the authors provide a table/chart that characterizes the study participants and a figure that shows the process of constructing the categories.

Response: Thank you, the above sections have been reviewed to address these comments, including a table characterizing the study participants.

Reviewer #2:

The qualitative design of this study is appropriate for the stated objectives, the methods (FGDs, KIIs, thematic analysis using NVivo, guided by the Bowen feasibility framework) are well aligned to the research questions, and the findings are clearly supported by the data presented. Inter-coder reliability checks were not included in the summary, so this would be one noted weakness.

Response: We thank the reviewer for the positive assessment of the study design, methodological alignment, and the clarity with which the findings are supported by the data.

We acknowledge the reviewer’s observation regarding the absence of an explicit description of inter-coder reliability checks in the initial summary. In response, we have revised the Methods section to clarify the analytic process, specifically noting that coding was conducted by multiple members of the research team using an iterative, team-based approach. Coding decisions were discussed regularly to compare interpretations, resolve discrepancies, and refine the coding framework.

Reviewer #3

Reviewer #3: This is an excellent addition to the G-ANC literature and should be valuable evidence to support policy changes in Tanzania. Overall the paper is well structured and well-written. It in some ways feels like 2 papers: 1 on experiences, and another on feasibility. It's good to keep them combined, but suggesting some re-structuring as well as other feedback:

Response: We thank the reviewer for the very positive assessment of the manuscript and for recognizing its contribution to the G-ANC evidence base and its relevance for policy in Tanzania.

We appreciate the observation that the manuscript may appear to address two strands (experiences and feasibility). We would like to clarify that the primary focus of the paper is on stakeholder experiences of G-ANC, with feasibility dimensions intentionally examined through the lens of these experiences..

In response to this feedback, we have revised the structure of the manuscript to more clearly foreground experiences and acceptability as the central analytical focus. Due to the length and focus of this paper, the constructs of feasibility and the this general term has been removed in this paper, instead some themes will be included in a separate manuscript which is in progress. We ensured that few feasibility-related insights are presented as experiential themes rather than as a parallel analytical stream. Revision of the Methods, Results and Discussion sections was done to improve narrative coherence.

1) It's helpful when research articles clearly state the study objectives being presented in the paper. In this case the results section covers 3 topics (e.g., experience, acceptability, feasibility) and each can be framed more explicitly as an objective. This would reshape the methods by adding a definition of acceptability, in addition to the feasibility definition and reference. The Bowen framework includes acceptability as a dimension of feasibility, but other Implementation Research frameworks (e.g., Proctor) define them separately. It would then follow the results section structure, and help restructure the Discussion around the 3 objectives. If the authors don't have enough additional content to add, then would suggest there be 2 explicit study objective on experience and on feasibility (with acceptability as one of the themes).

Response

We thank the reviewer for this thoughtful and constructive suggestion. We agree that clearly stated objectives strengthen coherence. In response, we have revised the manuscript to explicitly frame the manuscript around one objective:

1. To explore the experiences of women, healthcare providers, and health managers with the G-ANC) model.

we acknowledge that other implementation research frameworks (e.g., Proctor et al.) conceptualize acceptability as a distinct construct, we intentionally retained the Sekhon et al. for acceptability as it was most appropriate for this study.

Therefore we: Clarified the study objectives in the Introduction, Aligned the Methods section to reflect experience-focused and acceptability inquiry, revised the method section, results and discussion to follow this objective more explicitly presenting experience of stakeholders.

2) The paper can benefit from more G-ANC research context into the Background and Discussion so it's building on the full range of G-ANC in LMIC literature on the 2 outcomes (acceptability and feasibility), as well as the previous trial in Tanzania. There are many other studies that have reported on these 2 domains that can bring more context to these findings, and enhance the discussion by comparing what is similar or different from the Geita implementation. There are suggested citations as well as comments on the listed references for consideration. The background and Discussion could be revised to focus on those 2 outcomes, in addition to what's presented on experiences.

Response: Thank you for this important recommendation. In response, we have revised the manuscript incorporating additional published studies, Strengthen the Tanzania-specific context by referencing available published evidence relevant to group care, expanded the discussion by adding more comparative interpretation.

3) The description of participants throughout the paper is not always consistent which can be confusing, so this needs to be reviewed and streamlined from the abstract through the paper. Pregnant women and those who recently delivered were both FGD participants, but described differently in different sections. Also the description of providers (which cadres, and including CHWs who aren't typically called providers) and health managers can be explained in detail in the Methods section, and then terminology made consistent throughout.

Response: Thank you for highlighting this important issue. We agree that inconsistent terminology can be confusing for readers. We have reviewed and revised the manuscript to ensure consistent and clear description of participant groups. Specifically:

• We have standardized the terminology used for women participating in FGDs, consistently referring to them as pregnant and recently delivered women (within six months postpartum).

• We have clarified the composition of healthcare providers, only describing cadres involved (e.i., nurses, midwives, clinical officers and Doctors).

---

## [Editor Report · Decision Letter 1]

9 Mar 2026

Perspectives of women, healthcare providers and health managers on Group Antenatal Care implementation in Geita, Tanzania: A qualitative study

PONE-D-25-40423R1

Dear Dr. Kinyina,

We’re pleased to inform you that your manuscript has been judged scientifically suitable for publication and will be formally accepted for publication once it meets all outstanding technical requirements.

Kind regards,

Hannah Tappis, DrPH, MPH

Academic Editor

PLOS One

Additional Editor Comments (optional): All previous reviewer comments have been thoughtfully addressed.
---

## [Editor Report · Acceptance letter]

PONE-D-25-40423R1

PLOS One

Dear Dr. Kinyina,

I'm pleased to inform you that your manuscript has been deemed suitable for publication in PLOS One. Congratulations! Your manuscript is now being handed over to our production team.

Kind regards,

on behalf of

Dr. Hannah Tappis

Academic Editor

PLOS One